# Men's and women's knowledge of danger signs relevant to postnatal and neonatal care-seeking: A cross sectional study from Bungoma County, Kenya

Emma Roney[1,2], Christopher Morgan[1,2,3], Daniel Gatungu[4], Peter Mwaura[4], Humphrey Mwambeo[4], Alice Natecho[5], Liz Comrie-Thomson[1,3,6], Jesse N. Gitaka[4]*

1 Burnet Institute, Melbourne, Australia, 2 Melbourne School of Population and Global Health, University of Melbourne, Melbourne, Australia, 3 School of Public Health and Preventive Medicine, Monash University, Melbourne, Australia, 4 Directorate of Research and Innovation, Mount Kenya University, Thika, Kenya, 5 Fountain Africa Trust, Webuye, Bungoma, Kenya, 6 Department of Uro-Gynaecology, Ghent University, Ghent, Belgium

* jgitaka@mku.ac.ke

**Data Availability Statement:** The de-identified data underlying the results presented in the study are available from https://doi.org/10.6084/m9.figshare.

## Abstract

### Background

Neonatal and maternal mortality rates remain high in Kenya. Knowledge of neonatal danger signs may reduce delay in deciding to seek care. Evidence is emerging on the influential role of male partners in improving maternal and newborn health. This study analysed the factors that determine men's and women's knowledge and practices in postnatal and neonatal care-seeking, in order to inform design of future interventions.

### Methods

A quantitative, cross-sectional study was undertaken in Bungoma County, Kenya. Women who had recently given birth (n = 348) and men whose wives had recently given birth (n = 82) completed questionnaires on knowledge and care-seeking practices relating to the postnatal period. Univariate and multivariate logistic regression analyses were performed to investigate associations with key maternal and newborn health outcomes.

### Results

51.2% of women and 50.0% of men knew at least one neonatal danger sign, however women knew more individual danger signs than men. In the univariate model, women's knowledge of a least one neonatal danger sign was associated with attending antenatal care ≥4 times (OR 4.46, 95%CI 2.73–7.29, p<0.001), facility birth (OR 3.26, 95%CI 1.89–5.72, p<0.001), and having a male partner accompany them to antenatal care (OR 3.34, 95%CI 1.35–8.27, p = 0.009). Higher monthly household income (≥10,000KSh, approximately US$100) was associated with facility delivery (AOR 11.99, 95%CI 1.59–90.40, p = 0.009).

13048718. For researchers who meet the criteria for access to the confidential data can direct access requests to the Mount Kenya University Ethics Review Committee at research@mku.ac.ke.

**Funding:** HM AN and JG received funding through the Collaborative Newborn Support Project, which was supported by the UK Department for International Development's "Reducing Maternal and Neonatal Death in Kenya (2013-2018)" programme through the County Innovation Challenge Fund (grant number INN-R1-GA 004) (https://www.gov.uk/government/organisations/department-for-international-development). ER CM and LCT received no specific funding for this work. The funders had no role in study design, data collection and analysis, decision to publish, or preparation of the manuscript.

**Competing interests:** The authors have declared that no competing interests exist.

## Conclusion

Knowledge of neonatal danger signs was low, however there was an association between knowledge of danger signs and increased healthcare service use, including male partner involvement in antenatal care. Future interventions should consider the extra costs of facility delivery and the barriers to men participating in antenatal and postnatal care.

## Introduction

The 2030 Agenda for Sustainable Development emphasises the need to reduce preventable newborn deaths and the maternal mortality ratio [1]. Kenya's neonatal mortality rate (NMR) is 22 deaths per 1,000 live births, and Kenya's maternal mortality ratio (MMR) is 510 deaths per 100,000 live births [2]. This represents a major challenge if the Sustainable Development Goal targets of a NMR of 12 per 1,000 live births or lower, and an MMR of less than 70 per 100,000 live births [3], are to be met. Moreover, whilst worldwide under-5 child mortality has been declining, NMR has been reducing at a much slower rate, highlighting the need for urgent attention [4].

Low knowledge of obstetric and neonatal danger signs is widely reported throughout low- and middle-income countries [5–8]. JHPIEGO's birth preparedness and complication readiness framework proposes that increasing knowledge and awareness of these danger signs will improve problem identification, and thus will reduce the delay in deciding to seek care [9, 10]. Furthermore, caregivers' ability to recognise danger signs has been linked with use of antenatal care (ANC) and skilled birth attendance [7, 11, 12].

The importance of skilled birth attendance at every childbirth is widely recognised [11, 13], and has been described as the most significant single factor in averting maternal deaths [14]. Most maternal deaths occur at the time of labour, delivery and the immediate postpartum period [15], with 46% of all maternal deaths and 40% of all stillbirths and neonatal deaths occurring during the period of labour and the day of birth [16]. The importance of ANC during pregnancy is recognised in the World Health Organization's (WHO) revision of global ANC standards; from 2016 recommending eight ANC contacts throughout pregnancy [17]. However, in low-resource settings often less than half of women attend the pre-2016 recommendation of at least four ANC visits [8, 12, 18]. ANC is a crucial intervention in improving birth outcomes and reducing neonatal mortality [19], and is associated with increased likelihood of skilled delivery [20].

The Kenyan Government's commitment to improving maternal and newborn health outcomes is demonstrated by the 2013 Free Maternity Services Policy [21, 22], whereby maternal health services are delivered at no cost throughout the primary, secondary and tertiary government health sector [23]. Analysis of this policy shows that it has successfully started to increase facility deliveries, however challenges including knowledge and physical accessibility of the facilities still remain: it is clear that additional factors beyond cost affect uptake of facility-based care for pregnancy and childbirth [24]. Our research therefore seeks to understand the additional determinants of ANC use and childbirth occurring in a facility, especially those relating to women's and men's perspectives.

### Male involvement

Since the 1994 International Conference on Population and Development, there has been increasing recognition of the shared rights and responsibilities of women and men in sexual

and reproductive health, including the critical role of male partners in maternal and child health [3]. The WHO has listed male involvement as a key health promotion intervention for maternal and newborn health [25, 26]. It is recommended that men are engaged in health services and optimal home practices during pregnancy, childbirth and also after birth, however the level of evidence for interventions is low; resulting in calls for further research into the impacts of male involvement strategies on health outcomes [25, 27].

Strict definitions of what male involvement entails are not yet broadly agreed [27–29], however one elementary indicator is whether men participate in antenatal care visits [28]. The WHO recommendations stipulate that male involvement interventions ought to be culturally specific and thus may differ depending on the context [25, 26]. An important factor, however, is ensuring that male involvement interventions continue to promote, or at least not detract from, female autonomy and decision-making [25, 26, 30]. Pregnancy and childbirth are especially complex because these topics are often considered to be women's business [30], but men are often household decision makers in relation to care-seeking [27, 28].

## Collaborative Newborn Support Project

The 'Collaborative Newborn Support Project' has been implemented in Bungoma County, Kenya in order to reduce maternal and neonatal mortality from October 2015 to April 2019. It is an intervention of quasi-experimental design, involving newborn special care units, tele-health, call centre establishment, neonatology training, and community awareness programs [31]. The cross-sectional study reported here forms part of the 'Collaborative Newborn Support Project' and aimed to inform interventions through an assessment of men's and women's knowledge and practices relating to pregnancy and postnatal care.

## Methods

### Study design

This is a descriptive cross-sectional survey of women and men involved in the 'Collaborative Newborn Support Project', Kenya [31].

### Study setting

The study was undertaken in Bungoma County, Kenya. Bungoma County is located in Western Kenya, bordering Uganda, and has a population of 1.67 million people [32], mostly subsistence farmers [31], with only 11.3% of the population live in urban areas (2019 census) [33]. The project team believed it important to assess knowledge attitudes and practices of rural populations who were potential users of the hospitals in the broader project.

### Study population

Women and men from the same geographic area were recruited independently, meaning that the responses to each questionnaire are not linked as mother-father dyads. In total, 82 men and 348 women participated in the study, based on the data collection resources available to the project team. Men whose female partners had delivered within the previous one year (between April 2016 and April 2017) were recruited through convenience sampling, from those accompanying their female partners to healthcare clinics, and from men in market centres that fell within the regular catchment area of the facilities involved in the 'Collaborative Newborn Support Project', Kenya [31]. Women who had recently delivered were recruited at antenatal and postnatal reproductive care units, and in maternal and child health clinics at two health facilities: Bungoma and Webuye hospitals. These facilities were purposively sampled,

based on the fact that they are County and Sub-County referral hospitals respectively, and involved in the broader project intervention.

## Data collection

Two questionnaires were administered, one to the women's sample and the other to the men's sample. These were adapted and abridged from the JHPIEGO birth preparedness and complication readiness tool sample questionnaires [9]. Both questionnaires covered basic socio-demographic factors and asked similar questions about knowledge, attitudes and practices relating to maternal and newborn health; however, these weren't identical between the two surveys. Specific questions on danger signs in the postnatal period for woman or newborn classified these as Vaginal bleeding, Neo-natal sepsis, Jaundice, Convulsions, Asphyxia, High fever, Congenital problems, Difficulty breathing, Severe weakness, Changed activity, Bleeding umbilical cord, Poor breastfeeding, and free-text fields for other options. Data were collected by research assistants over a three-month period in 2017. The questionnaires were in English and Kiswahili. Research assistants translated the questions into local dialect whenever necessary. Following collection, the data were transferred to an Access database and archived in Mount Kenya University servers within the Directorate of Research and Innovation.

## Data analysis

The data were cleaned and analysed using Stata 13 [34] to find summary statistics and to undertake univariate and multivariate logistic regression analyses. Complete case analysis was used in regression analyses. Some variables were grouped to dichotomous responses, based on analysis team consensus, to ensure no group was too small for regression analysis. The common approach of interpreting a p-value of less than 0.05 as indicating statistical significance was taken.

Although questions were not completely uniform across the two questionnaires, where possible we used similar variables across both data sets in the analysis to allow the contrast of women's and men's knowledge and practices. Outcome variables examined in the logistic regression analysis covered both knowledge and practices. There was insufficient variability in the data to include attitudes in the final analysis.

Univariate associations were tested between outcome variables and hypothesised factors of association, as determined by similar studies and the availability of data from the questionnaires. These were then included in a multivariate model to control for the effects of confounding. Potential confounding factors included in the women's multivariate model were women's and men's age and education levels, monthly household income, time to healthcare facility, gravidity, age at first pregnancy and shared decision making for health service seeking between woman and male partner. Potential confounding factors included in the men's multivariate model were women's and men's age and education level, and monthly household income. Outcomes considered to be on the causal pathway between exposure and outcome were included in univariate models, but not in multivariate models.

## Ethical considerations

This study was approved by the Mount Kenya University Ethics Review Committee (MKU/ERC/0096). Participants signed informed consent forms after the aims and research process were explained to them, prior to undertaking the questionnaire.

## Results

### Socio-demographic characteristics

**Women's sample.** Half of women were under 25 years of age (49.4%) with most of their husbands at least 30 years of age (67%). Over a third were first time mothers (38.2%) and around half had completed secondary school or higher (54.8%). Women reported their husbands as slightly higher educated, with 68.7% having completed secondary school or higher (at least 12 years of formal school). 63.3% lived in households with a monthly income at or above 10,000 KSh (approximately USD100). More than half (59.1%) lived more than 5 kilometres away from the nearest health facility. Over a third of women reported sharing pregnancy and childbirth decision making with their husband (38.8%), with the rest either making the decision themselves, their husband making the decision without them, or the decision was made by another family member, such as a mother in law (Table 1).

**Men's sample.** Since the women's and men's cohorts were not recruited in the same way, they represent comparable, but not matched, populations. Sampled male respondents, along with their wives, tended to be slightly older, more educated and in higher income households than those in the women's sample. Most men were 30 years of age or older (71.9%), with most of their wives at least 25 years of age (69.5%). Three quarters of the men had completed secondary school or higher (75.6%), while 65.8% of their wives had completed secondary school or higher. Most households had a monthly income of over 10,000 KSh (80.5%) (Table 1).

### Women's and men's knowledge and practices relating to pregnancy and postnatal care

Just over half of the women (51.2%) knew at least one neonatal danger sign, with over a third of women reporting that their newborn had experienced problems after childbirth (39.1%). Half of those women who had experienced a problem sought care within an hour of noticing the newborn was ill (53.2%), however, 30.6% waited for more than six hours before seeking care. Regarding antenatal care, 39.1% of women attended at least four times during the last pregnancy. Most women delivered in a healthcare facility (79.2%), be that a government hospital or a private clinic. In the men's sample, 40.2% of the men knew at least one postpartum danger sign and half (50.0%) knew at least one neonatal danger sign. 51.2% of men accompanied their wife to antenatal care during her most recent pregnancy, either always, most of the time or sometimes. Almost all men reported that their wife's last childbirth was in a healthcare facility (90.1%), with two thirds accompanying their wife to health facility for delivery (66.2%). Most deliveries occurred in a government hospital (79.0%) and were attended by a nurse or midwife (73.8%) (S1 Table).

**Identification of neonatal danger signs.** Fig 1 illustrates that overall the sample of women could identify a larger number of danger signs than the men, although a similar proportion of women and men were not able to identify any neonatal danger signs (48.8% of women, compared to 50.0% of men). Each individual danger sign was identified by a greater percentage of women than men. Some danger signs were identified approximately twice as frequently by women than by men, such as poor breastfeeding or not able to breastfeed, fast breathing, fever, and difficult to wake, lethargic or unconscious.

### Characteristics associated with knowledge of neonatal danger signs

**Characteristics associated with women's knowledge.** Higher levels of woman's education (AOR: 5.65, 95% CI: 1.88–17.04, p = 0.002), higher household income (AOR: 4.35, 95% CI: 1.74–10.88, p = 0.002), multigravidity (AOR: 4.66, 95% CI: 1.52–14.36, p = 0.007), and

**Table 1. Socio-demographic characteristics of the women's and men's samples.**

| | WOMEN | | MEN | |
|---|---|---|---|---|
| | n = 348 | | n = 82 | |
| | N | % | N | % |
| Woman's age (years) | | | | |
| <25 | 172 | 49.4 | 25 | 30.5 |
| ≥25 | 176 | 50.6 | 57 | 69.5 |
| Woman's median age | | 25 years | | 28 years |
| Man's age (years) | | | | |
| <30 | 74 | 33.0 | 23 | 28.1 |
| ≥30 | 150 | 67.0 | 59 | 71.9 |
| Man's median age | | 30.5 years | | 33.5 years |
| Woman's highest level of education completed | | | | |
| Primary school or less | 157 | 45.2 | 28 | 34.2 |
| Secondary school or greater | 191 | 54.8 | 54 | 65.8 |
| Man's highest education level completed | | | | |
| Primary school or less | 74 | 31.3 | 20 | 24.4 |
| Secondary school or greater | 163 | 68.7 | 62 | 75.6 |
| Monthly household income (KSh) | | | | |
| <10,000 | 79 | 36.7 | 16 | 19.5 |
| ≥10,000 | 136 | 63.3 | 66 | 80.5 |
| Distance to healthcare facility from home | | | | |
| ≤5 kilometres | 141 | 40.9 | | |
| >5 kilometres | 204 | 59.1 | | |
| Gravidity | | | | |
| Primigravida | 129 | 38.8 | | |
| Multigravida | 209 | 61.2 | | |
| Age at first pregnancy | | | | |
| <18 | 85 | 25.0 | | |
| ≥18 | 255 | 75.0 | | |
| Shared decision making for health service seeking between mother and male partner | | | | |
| No | 211 | 61.2 | | |
| Yes | 134 | 38.8 | | |

* Note that the women's and men's cohorts were not recruited in the same way and are therefore not entirely comparable.

older age at first pregnancy (AOR: 3.24, 95% CI: 1.00–10.52, p = 0.05) were all significantly associated with woman's knowledge of at least one neonatal danger sign, after adjusting for confounding (S2 Table).

**Characteristics associated with men's knowledge.** Higher household income, at or above KShs 10,000 per month, was significantly associated with men's knowledge of at least one neonatal danger sign (AOR: 4.09, 95%CI: 1.00–16.64, p = 0.049) (S3 Table).

## Characteristics associated with health care-seeking practices

**Characteristics associated with women's care-seeking.** Women who shared care-seeking decision making with their husband had increased odds of attending antenatal care at least four times throughout their most recent pregnancy (AOR: 2.27, 95% CI: 1.10–4.67, p = 0.027). Women who knew at least one neonatal danger sign had increased odds of attending four or

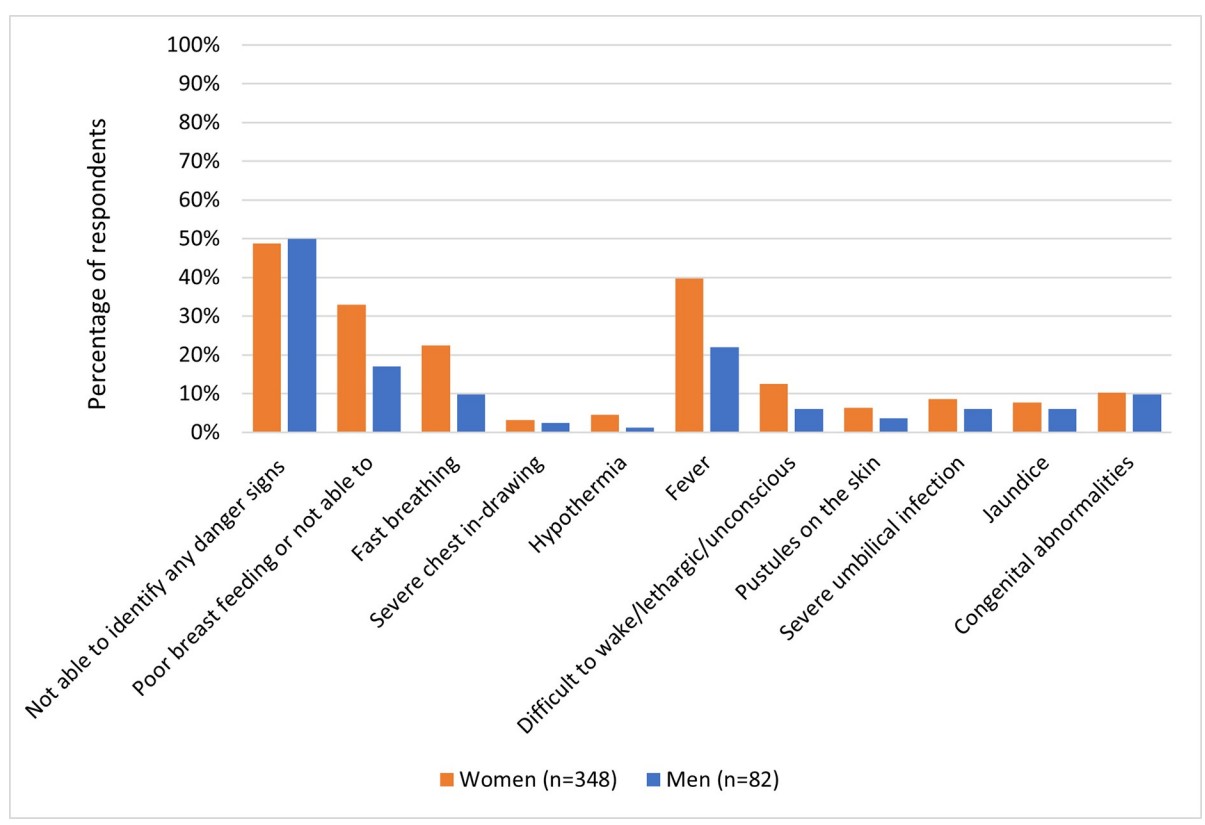

**Fig 1. Identification of neonatal danger signs: Women and men**∗.

more antenatal visits during their last pregnancy, compared with those who did not know any neonatal danger signs (unadjusted OR: 4.46, 95% CI: 2.73–7.29, p<0.001) (S4 Table).

Women who completed secondary school or higher had increased odds of birth in a healthcare facility, compared with those who had completed primary school or less, after adjusting for confounding (AOR: 4.24, 95% CI: 1.35–13.30, p = 0.013). The univariate analysis for those factors on the causal pathway also demonstrated that there was a significant increase in the odds of the most recent birth occurring in a healthcare facility both for women who knew at least one neonatal danger sign (OR: 3.26, 95% CI: 1.89–5.72, p<0.001), and for women who attended antenatal care at least four times during their last pregnancy (OR: 5.20, 95% CI: 2.38–11.39, p<0.001) (S4 Table).

**Characteristics associated with men's practices.** In the men's sample, both male partner's and woman's education was associated with men accompanying their wife to antenatal care during her most recent pregnancy. However, these associations were in opposing directions. Men who completed secondary school or higher had an odds ratio of 0.25 (95% CI: 0.07–0.95, p = 0.042), meaning they had a 75% decrease in odds of accompanying their wife to antenatal care, compared with counterparts who had completed primary school only. Conversely, men whose wife had completed secondary school or higher had increased odds of accompanying their wife to antenatal care (AOR: 3.45, 95% CI: 1.09–11.28, p = 0.036). In univariate analysis, men who knew at least one neonatal danger sign showed increased odds of accompanying their wife to antenatal care during her most recent pregnancy (AOR: 3.34, 95% CI: 1.35–8.27, p = 0.009) (S5 Table).

Men in households with a monthly income at or above KShs 10,000 had increased odds of their wives delivering in a healthcare facility (AOR: 11.99, 95% CI: 1.59–90.40, p = 0.016), compared with those who were in households with monthly incomes below 10,000 KSh, after adjusting for confounding factors (S6 Table).

## Discussion

Overall, knowledge of neonatal danger signs was low, a finding mirrored in similar studies in other low- and middle-income settings, as well as elsewhere in Kenya [6, 7, 35, 36]. Women were better able than men to identify danger signs, especially when asked to name specific danger signs, even though the men's sample tended to be older, more educated, and from higher income households. In general, knowledge of danger signs was associated with wealth and education, as others have also found [7, 35–37]. If men are to be actively involved in decision making and healthcare seeking in maternal and newborn health, improving their knowledge of key issues such as danger signs is a necessary starting point. Our regression models show strong associations between women's and men's knowledge of neonatal danger signs and positive healthcare seeking behaviours in pregnancy and postnatal care. Whilst the cross-sectional study design of this research cannot show causation, these associations suggest a correlation between danger signs knowledge and healthcare seeking practices during pregnancy and childbirth.

The results show that knowledge is not adequate in explaining delays in health care seeking among mothers of newborns: 30% of women waited over 6 hours after recognising that their newborn was ill before seeking care. Other factors observed to influence health care-seeking in our results echo the findings of others in suggesting that education and wealth are key determining factors in use of antenatal care [18], and skilled delivery [12, 21, 35]. In both the men's and women's samples, higher income was associated with both higher knowledge and healthier practices, such as a woman's most recent childbirth occurring in a facility, even after adjusting for confounding factors. Given that Kenya now has free maternity health services [22, 23], there may be other cost barriers, aside from cost of care itself, at play in the decision to seek care. Future programs may need to consider, for example, the cost of transport, accommodation, and the opportunity cost of missing employment. Our age and parity findings also suggest that health promotion interventions and health communication initiatives should target first time and/or young mothers; this may also present an opportunity for peer-based community learning, such as through group antenatal care [38], whereby more experienced and older mothers are able to assist in the teaching of danger signs.

This study has highlighted the low proportion of men accompanying their wives to antenatal care, as seen elsewhere [39]. The conflicting findings whereby more educated women had healthier practices, but more educated male partners (in the men's sample) seemed less likely to accompany their wives to antenatal care warrants consideration. It may be that better educated men are more likely to be in formal employment, and programs aiming to increase male involvement may need to consider clinic scheduling that better enables men (and women) to balance parental responsibilities with paid work requirements, for example by offering ANC sessions on weekends, outside office hours, or close to the workplace. Shifting gender norms and attitudes among men, women, health providers and employers can also be expected to contribute to men's increasing participation in antenatal care, for example by reducing stigma, normalising men's leave from work or flexible working, and providing inclusive antenatal health services that address both parents' reproductive health needs. Further research, particularly qualitative research, on the challenges and barriers men face in attending antenatal care in Bungoma County would prove useful for informing future program design. For example,

other studies in different locations comment on the ridicule men can face upon their arrival at maternal and child health services [35, 40], and also the lack of infrastructure to accommodate men's presence, such as privacy curtains in clinics [41]. Additionally, there is opportunity in enhancing utility of the Mother Child Handbook that details pregnancy and neonatal danger signs by targeting communication to men on using the handbook to increase their knowledge on the danger signs. Presently the handbook is more mother focussed.

## Limitations and strengths

A key limitation of this research is the small sample size of the men's sample, at 82 participants. This meant that confidence intervals were wide and there was often weak evidence of associations since the small sample size reduced the power of the study to detect smaller differences. As a convenience sample, these results must be taken as indicative of potential knowledge patterns rather than being representative of the general male population. Due to limitations in the project's data collection resources, women and men were also administered different questionnaires and were sampled in different ways. This limits the potential for direct comparison between the two groups, and because some questions were worded slightly differently for men and women, no statistical analysis of knowledge differences was possible. We estimated that we were going to get less biased cohort of men in market places than the few motivated ones who accompany their partners in the ante natal clinics. Additionally, due to cultural barriers, men who accompany their partners to ANC are really outliers. Relying exclusively on this catchment may have denied the study representativeness of the general population. The regression analysis undertaken used similar outcomes and exposures in order to smooth the differences between the two cohorts.

The primary strength of this research is that it was conceived and led by local Kenyan researchers and thus addressed local priorities. Additionally, this research has combined data from both women and men in one study, which is not widely seen in the existing literature, thus enabling a level of comparison between these two interconnected groups.

## Conclusion

Overall, knowledge of neonatal danger signs in this particular population within Bungoma County, Kenya is low among both women and men. Whilst it is suggested that improving knowledge of the neonatal danger signs can reduce the delay in deciding to seek care [9, 10], there still exists a certain disconnect between knowledge translating into practice for some women in Bungoma County who did not immediately seek care once realising their newborn was ill.

The key determining factors in men's and women's knowledge and practices relating to pregnancy and postnatal care were education level, income, gravidity and age at first pregnancy. Future interventions, including those in the 'Collaborative Newborn Support Project', must thus consider the extra costs of childbirth occurring in a facility. Furthermore, interventions should address the barriers to men participating in antenatal care, including work commitments and pervasive social and gender norms around pregnancy and child-rearing. Target groups for knowledge-based interventions should focus on primiparous mothers and mothers under the age of 18 at their first pregnancy. These findings have important implications those working in maternal and newborn health in Bungoma County, as well as the overall reduction of maternal and neonatal mortality in Kenya.

## Supporting information

**S1 Table. Women's and men's knowledge and practices relating to pregnancy and postnatal care.**
(DOCX)

**S2 Table. Factors associated with women's knowledge of at least one neonatal danger sign.**
(DOCX)

**S3 Table. Factors associated with men's knowledge of at least one neonatal danger sign.**
(DOCX)

**S4 Table. Factors associated with women's healthy care-seeking practices.**
(DOCX)

**S5 Table. Factors associated with male partners accompanying women to antenatal care during her most recent pregnancy.**
(DOCX)

**S6 Table. Factors associated with a woman's most recent birth occurring in a healthcare facility, as reported by the male partner.**
(DOCX)

## Acknowledgments

All authors gratefully acknowledge the participation of the men and women who contributed data to this study. Burnet Institute acknowledges the contribution to this work of the Victorian Operational Infrastructure Support Program.

## Author Contributions

**Conceptualization:** Jesse N. Gitaka.

**Formal analysis:** Emma Roney, Christopher Morgan.

**Methodology:** Daniel Gatungu, Humphrey Mwambeo, Alice Natecho, Jesse N. Gitaka.

**Project administration:** Daniel Gatungu, Peter Mwaura, Humphrey Mwambeo, Alice Natecho, Jesse N. Gitaka.

**Validation:** Liz Comrie-Thomson.

**Writing – original draft:** Emma Roney, Christopher Morgan.

**Writing – review & editing:** Emma Roney, Christopher Morgan, Daniel Gatungu, Peter Mwaura, Humphrey Mwambeo, Alice Natecho, Liz Comrie-Thomson, Jesse N. Gitaka.

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
