## [Decision Letter · Decision Letter 0]

5 May 2020

PONE-D-20-06532

Men’s and women’s knowledge and practices relating to pregnancy and postnatal care: a cross sectional study from Bungoma County, Kenya

PLOS ONE

Dear Dr. Gitaka,

Thank you for submitting your manuscript to PLOS ONE. After careful consideration, we feel that it has merit but does not fully meet PLOS ONE’s publication criteria as it currently stands. Therefore, we invite you to submit a revised version of the manuscript that addresses the points raised during the review process.

We would appreciate receiving your revised manuscript by Jun 19 2020 11:59PM. To enhance the reproducibility of your results, we recommend that if applicable you deposit your laboratory protocols in protocols.io, where a protocol can be assigned its own identifier (DOI) such that it can be cited independently in the future. For instructions see: http://journals.plos.org/plosone/s/submission-guidelines#loc-laboratory-protocols

We look forward to receiving your revised manuscript.

Kind regards,

Emma Sacks

Academic Editor

PLOS ONE

2. In your Methods section, please provide additional information about the participant recruitment method and the demographic details of your participants for both the womens and mens samples. Please ensure you have provided sufficient details to replicate the analyses such as: a) the recruitment date range (month and year), b) a description of any inclusion/exclusion criteria that were applied to participant recruitment, c) a table of relevant demographic details, d) a statement as to whether your sample can be considered representative of a larger population, e) a description of how participants were recruited, and f) descriptions of where participants were recruited and where the research took place."

3. Please provide a sample size and power calculation in the Methods, or discuss the reasons for not performing one before study initiation."  

4. Please include additional information regarding the survey or questionnaire used in the study and ensure that you have provided sufficient details that others could replicate the analyses. For instance, if you developed a questionnaire as part of this study and it is not under a copyright more restrictive than CC-BY, please include a copy, in both the original language and English, as Supporting Information.

6. Please include your tables as part of your main manuscript and remove the individual files. Please note that supplementary tables (should remain/ be uploaded) as separate "supporting information" files

Editor Comments:

Thank you for your revisions. However, there are still many comments from reviewers to address, but we are confident they will make the article stronger.

In addition to those from the reviewer:

-There is inconsistency is the statement of the objective. The title states knowledge, but the paper suggests knowledge, attitudes and practices, and also at times says "determinants of ANC use and facility based childbirth. Please clarify.

-Further, the link between knowledge and male involvement isn't very clear.

-The data presented are not very in-depth and it is not clear what this adds to previous literature on this topic, even from Kenya, since there are existing studies. Some of the more nuanced analysis may be in supplemental tables and the authors may want to consider including those info in the main paper.

-There is some inconsistency in language about the questionnaire - in some places it is referred to as an interview

-I wouldn't say that fathers being influential in maternal and child health is new

-"recently" delivered needs to be defined (was it deliveries within a year?); also, how were recently delivered women recruited from ANC??

-the association between knowledge of danger signs and ANC is presented as if knowledge is a factor related to ANC, when it is likely that the association is the other way (those who attend ANC or deliver in a facility learn the danger signs)

-low knowledge of danger signs is not only in low income countries, but also among those who are disadvantaged/uneducated in high income settings

-if the authors refer to the "first delay" then the 3 delays model has to be explained

-how many danger signs were included in the questionnaire? what were they?

-For the languages, it if was expected to need other languages besides English and Kiswahili, why not translate ahead of time? it is atypical to use ad hoc translation, as the terminology is then not consistent.

-there is redundant demographic info in the descriptions of the male and female samples

-there should not be new findings presented in the discussion. The results around how long it took to seek care once identifying an illness should be in the results.

-the limitations allude to confidence intervals but these are not presented

-why were the women's and men's cohorts not recruited the same way? Needs to be explained in the limitations.

Reviewers' comments:

Reviewer's Responses to Questions

**Comments to the Author**

1. Is the manuscript technically sound, and do the data support the conclusions?

Reviewer #1: Yes

Reviewer #2: Yes

Reviewer #3: Yes

2. Has the statistical analysis been performed appropriately and rigorously? 

Reviewer #1: Yes

Reviewer #2: Yes

Reviewer #3: Yes

3. Have the authors made all data underlying the findings in their manuscript fully available?

Reviewer #1: Yes

Reviewer #2: Yes

Reviewer #3: Yes

4. Is the manuscript presented in an intelligible fashion and written in standard English?

Reviewer #1: Yes

Reviewer #2: Yes

Reviewer #3: Yes

5. Review Comments to the Author

Reviewer #1: Thank you for the opportunity to review this manuscript, which tackles the important problem of parents knowledge about neonatal danger signs using a cross sectional study from Bungoma County, Kenya.

Is the title reflective of the main message: Men’s and women’s knowledge and practices relating to pregnancy and postnatal care—isn’t it really focused on neonatal danger signs?

This is a strikingly well written manuscript which has been meticulously copy edited and is easy to read and follow. It makes clear points.

I have a few comments which I will hope to strengthen the manuscript further:

1. In the abstract, why are unadjusted models of mothers’ knowledge presented instead of adjusted ones like for the men?

2. Similarly, in the body of the manuscript in the results section please focus only on the adjusted odds ratio’s and remove the unadjusted ones. Flipping back-and-forth between adjusted and unadjusted ones is confusing to the reader

3. Why do the conclusions in the abstract only refer to the mothers’ knowledge?

4. Please clarify in the introduction when ANC is used whether you mean all of antenatal care or antenatal and intrapartum and postpartum care e.g. particularly in lines 81 - 89, but also elsewhere.

5. Another key limitation is that we do not know the response rate

6. In the limitations or strengths could you please mention the degree to which your sample is reflective of the general population in terms of education, age, etc.

7. For your tables, rather than putting the 1.00 in each of the cells indicating the reference, would you consider decluttering the table by putting (Reference) in the first column only? eg Less than 25 years (Reference)

8. Similarly, since you were 95% confidence intervals indicate significance you could delete the P values and just bold the odds ratio’s and 95% confidence intervals instead which would decrease clutter within the table and make it easier for your readers to see the main points

9. Throughout the tables, figures, and manuscript rather than using “father“ please replace with “men” as you do in figure 1 for example. Why? We know from genetics research that 10% of male partners are not actually the father of the child

10. Please comment on missing data for each question

11. Please clarify how missing data was handled e.g. ?complete case analysis

Minor points:

Please do not start sentences with numbers e.g. line 190

When you print off your figures in black-and-white, are the two colours easily distinguishable?

Reviewer #2: I have made separate comments that need to be addressed. These are scattered through the document so can not be displayed here. The attachment will show the areas needing review by the authors. It is a worthwhile papers that shows where health workers need to focuss on

Reviewer #3: The methodology used should be justified. For example, why is it that male partners of female participants were not enrolled, instead men we conveniently enrolled from the market place. What was the basis of sample size calculation? Why the difference in sample size? What is the role of the collaborative intervention in this study? The clarity of the methodology will help in the clarity of the results, discussion and conclusions. More comments are in the document.

6. PLOS authors have the option to publish the peer review history of their article (what does this mean?). If published, this will include your full peer review and any attached files.

Reviewer #1: Yes: Sarah McDonald

Reviewer #2: No

Reviewer #3: No

---

## [Author Response · Author response to Decision Letter 0]

1 Nov 2020

 Done

 2. In your Methods section, please provide additional information about the participant recruitment method and the demographic details of your participants for both the womens and mens samples. Please ensure you have provided sufficient details to replicate the analyses such as: a) the recruitment date range (month and year), b) a description of any inclusion/exclusion criteria that were applied to participant recruitment, c) a table of relevant demographic details, d) a statement as to whether your sample can be considered representative of a larger population, e) a description of how participants were recruited, and f) descriptions of where participants were recruited and where the research took place."

Done. See the methods and results sections as well as the supplementary tables.

3. Please provide a sample size and power calculation in the Methods, or discuss the reasons for not performing one before study initiation." 

Women and men from the same geographic area were recruited independently, meaning that the responses to each questionnaire are not linked as mother-father dyads. In total, 82 men and 348 women participated in the study. Men whose female partners had recently delivered were recruited through convenience sampling, from those accompanying their female partners to healthcare clinics, and from men in market centres. Women who had recently delivered were recruited at antenatal and postnatal reproductive care units, and in maternal and child health clinics at two health facilities of Webuye level 5 hospital and Bungoma Referral hospital

4. Please include additional information regarding the survey or questionnaire used in the study and ensure that you have provided sufficient details that others could replicate the analyses. For instance, if you developed a questionnaire as part of this study and it is not under a copyright more restrictive than CC-BY, please include a copy, in both the original language and English, as Supporting Information.

 Copies provided

We have shared de-identified data at https://doi.org/10.6084/m9.figshare.13048718. Contact information for Mount Kenya University Ethics Review Committee is research@mku.ac.ke.

 We have uploaded the 2 datasets at https://doi.org/10.6084/m9.figshare.13048718.

6. Please include your tables as part of your main manuscript and remove the individual files. Please note that supplementary tables (should remain/ be uploaded) as separate "supporting information" files

 Done

 Done

Editor Comments:

Thank you for your revisions. However, there are still many comments from reviewers to address, but we are confident they will make the article stronger.

In addition to those from the reviewer:

-There is inconsistency is the statement of the objective. The title states knowledge, but the paper suggests knowledge, attitudes and practices, and also at times says "determinants of ANC use and facility based childbirth. Please clarify.

The new title captures the attitudes as well.

-Further, the link between knowledge and male involvement isn't very clear.

Indeed this forms part of our research question; what is the best way to address male involvement? What are the knowledge gaps for male involvement?

-The data presented are not very in-depth and it is not clear what this adds to previous literature on this topic, even from Kenya, since there are existing studies. Some of the more nuanced analysis may be in supplemental tables and the authors may want to consider including those info in the main paper.

We have included more tables to deepen the analyses. Our study is unique in that we address contextual issues in a high burden setting. 

-There is some inconsistency in language about the questionnaire - in some places it is referred to as an interview.

Corrected.

-I wouldn't say that fathers being influential in maternal and child health is new.

Indeed, this is not new. However we aimed at understanding contextual issues around male involvement in knowledge and practice. In the study context, men have traditionally left child care to women. This perception has been so pervasive that men have been left out in the design of maternal and child health programs and policies. This study hoped to establish a scientific basis for either supporting or refuting these perceptions.

-"recently" delivered needs to be defined (was it deliveries within a year?); also, how were recently delivered women recruited from ANC??

Recently delivered means deliveries within a year. Yes, these were recruited from the ANC.

-the association between knowledge of danger signs and ANC is presented as if knowledge is a factor related to ANC, when it is likely that the association is the other way (those who attend ANC or deliver in a facility learn the danger signs)

This is well understood. However, in our setting, the extended family is also an important source of knowledge on danger signs.

-low knowledge of danger signs is not only in low income countries, but also among those who are disadvantaged/uneducated in high income settings

This is true, but also outside of the scope of this paper, which focuses on Kenya.

-if the authors refer to the "first delay" then the 3 delays model has to be explained

We have taken out the word “first” so as to not complicate things with long explanations of a well-known concept. 

-how many danger signs were included in the questionnaire? what were they?

Vaginal bleeding

Neo-natal sepsis

Jaundice

Convulsions

Asphyxia

High fever

Congenital problems

Difficulty breathing

Severe weakness

Accelerated/reduced foetal movement

Water breaks without labour

Bleeding umbilical cord

Poor breastfeeding

Other- open ended.

-For the languages, it if was expected to need other languages besides English and Kiswahili, why not translate ahead of time? it is atypical to use ad hoc translation, as the terminology is then not consistent.

Most of the participants understand Kiswahili, and hence we estimated that there was minimal need to translate to Kibukusu or kisabaoti.

-there is redundant demographic info in the descriptions of the male and female samples

Corrected.

-there should not be new findings presented in the discussion. The results around how long it took to seek care once identifying an illness should be in the results.

This information is presented in both the results table S1 and the body of the text (currently line 289)

-the limitations allude to confidence intervals but these are not presented.

These are in the supplementary tables.

-why were the women's and men's cohorts not recruited the same way? Needs to be explained in the limitations.

Done.

Reviewers' comments:

Reviewer's Responses to Questions

Comments to the Author

Reviewer #1: Thank you for the opportunity to review this manuscript, which tackles the important problem of parents knowledge about neonatal danger signs using a cross sectional study from Bungoma County, Kenya.

Is the title reflective of the main message: Men’s and women’s knowledge and practices relating to pregnancy and postnatal care—isn’t it really focused on neonatal danger signs?

Well noted and corrected.

This is a strikingly well written manuscript which has been meticulously copy edited and is easy to read and follow. It makes clear points.

I have a few comments which I will hope to strengthen the manuscript further:

1. In the abstract, why are unadjusted models of mothers’ knowledge presented instead of adjusted ones like for the men?

2. Similarly, in the body of the manuscript in the results section please focus only on the adjusted odds ratio’s and remove the unadjusted ones. Flipping back-and-forth between adjusted and unadjusted ones is confusing to the reader.

We reported on unadjusted odds ratios when the variable was considered to be on the causal pathway, and therefore not included in the multivariate model. This was explained in the methods and is noted in the relevant tables. If we take out the unadjusted ORs, we will lose a big chunk of the data/paper content.

3. Why do the conclusions in the abstract only refer to the mothers’ knowledge?

We have states that “knowledge of neonatal danger signs was low”, which was true for both mothers and men.

4. Please clarify in the introduction when ANC is used whether you mean all of antenatal care or antenatal and intrapartum and postpartum care e.g. particularly in lines 81 - 89, but also elsewhere.

We have clarified that this is during pregnancy.

5. Another key limitation is that we do not know the response rate

The response rate was 88%.

6. In the limitations or strengths could you please mention the degree to which your sample is reflective of the general population in terms of education, age, etc.

Done.

7. For your tables, rather than putting the 1.00 in each of the cells indicating the reference, would you consider decluttering the table by putting (Reference) in the first column only? eg Less than 25 years (Reference)

Done.

8. Similarly, since you were 95% confidence intervals indicate significance you could delete the P values and just bold the odds ratio’s and 95% confidence intervals instead which would decrease clutter within the table and make it easier for your readers to see the main points

We wish to maintain as is, as these complement each other and is the practice generally.

9. Throughout the tables, figures, and manuscript rather than using “father“ please replace with “men” as you do in figure 1 for example. Why? We know from genetics research that 10% of male partners are not actually the father of the child

Done.

10. Please comment on missing data for each question.

We had very few missing data which were treated automatically under regression analyses in stata.

11. Please clarify how missing data was handled e.g. ?complete case analysis

Done.

Minor points:

Please do not start sentences with numbers e.g. line 190

Well noted and corrected.

When you print off your figures in black-and-white, are the two colours easily distinguishable?

Yes.

Reviewer #2: I have made separate comments that need to be addressed. These are scattered through the document so can not be displayed here. The attachment will show the areas needing review by the authors. It is a worthwhile papers that shows where health workers need to focuss on

Reviewer #3: The methodology used should be justified. For example, why is it that male partners of female participants were not enrolled, instead men we conveniently enrolled from the market place. What was the basis of sample size calculation? Why the difference in sample size? What is the role of the collaborative intervention in this study? The clarity of the methodology will help in the clarity of the results, discussion and conclusions. More comments are in the document.

 Done. See methods section.

 Reviewer 2 extended comments: 

Comment: The report is centred on knowledge of newborn danger sign. This should be reflected in the title ie replace “relating to pregnancy and postnatal care”. There are no results on postnatal care

Well noted and done.

ABSTRACT: The authors studied knowledge rather than costs of care. Further studied should address knowledge

Agreed and corrected. Thank you.

INTRODUCTION

Lines 87-89 & 116-117 

“Our research therefore seeks to understand the additional determinants of ANC use and childbirth occurring in a facility, especially those relating to mothers’ and fathers’ perspectives.

The cross-sectional study reported here forms part of the ‘Collaborative Newborn Support Project’ and aimed to inform interventions through an assessment of men’s and women’s knowledge and practices relating to pregnancy and postnatal care.”

Comment: It’s difficult to correlate ‘additional determinants of ANC use and childbirth’ ‘through an assessment of men’s and women’s knowledge and practices’

Indeed the current study there is little information on actual determinant of use but the authors mostly report on determinants of knowledge.

ANC use is a practice, so we are not being contradictory here. We look at determinants of knowledge and then if knowledge is a determinant of use/practice.

Also give the readers a brief on the broader ‘Collaborative Newborn Support Project’ how long has been going and at stage was the current done.

The Collaborative Newborn Support Project ran from October 2015 to April 2019 and the current study was undertaken July to December 2017.

METHODS

Study setting does not include the hospitals (for women & men) and the type of markets (for men) where the participants were recruited from.

What was their definition of 1. “recently delivered” 2. “delay in seeking health care for the baby?

1. Recently delivered- means had a delivery in the last 1 year.

2. Delay in seeking health care for the baby- Time taken after recognition for need to seek care. We classified this in 3 time bands <1 hour, 1-6 hours and >6 hours.

RESULTS

Socio-demographic characteristics &

 Table 1

• To avoid confusion in the column for each characteristic, use men or women instead of husband’s or mother’s. 

Done

• Give median ages for the women & their partners in both columns.

Done

Do analyses for all the observations to show if there was any significant statistical differences 

We intentionally did not do this because the data are actually not the same.

Men’s sample: In the narrative give the proportion of men who had accompanied their partners in the sample

This is already included under the heading ‘women’s and men’s knowledge and practices relating to pregnancy and postnatal care’ (currently line 292-293)

Supplementary materials

Table S1

Knowledge of at least one postpartum danger sign – which one?

‘’At least one danger sign” refers to respondents listing one or more of the danger signs in their response and does not limit them to a particular one.

The authors need to explain why for women data was not reported on the following:

1. Knowledge of at least one postpartum danger sign

2. Accompanied by partner to ANC

3. Accompanied by partner to delivery 

These weren’t questions in the women’s questionnaire. We do say in the methods that the questionnaires were similar. We have added in slightly more explanation to hopefully make this clearer.

DISCUSSION

Kenya has a “Mother Child Health Handbook” which is given for each pregnancy and is in taken home by the mother. This handbook has nearly all the information these authors were looking for in the current study, but it is not mentioned at all in the discussion of this paper 

 The Mother Child Handbook captures most of the information we sought, however, in our study we aimed at studying the knowledge, attitudes and practices in our setting in Bungoma County where maternal and newborn indicators are among the worst in Kenya. In revision, we have highlighted the need to enhance utility of the Mother Child Handbook as a source of information on danger signs for families.

---

## [Decision Letter · Decision Letter 1]

7 Dec 2020

PONE-D-20-06532R1

Men’s and women’s knowledge, attitudes and practices relating to pregnancy and neonatal care: a cross sectional study from Bungoma County, Kenya

PLOS ONE

Dear Dr. Gitaka,

Thank you for submitting your manuscript to PLOS ONE. After careful consideration, we feel that it has merit but does not fully meet PLOS ONE’s publication criteria as it currently stands. Therefore, we invite you to submit a revised version of the manuscript that addresses the points raised during the review process.

We look forward to receiving your revised manuscript.

Kind regards,

Emma Sacks

Academic Editor

PLOS ONE

Additional Editor Comments (if provided):

Thank you for these edits - the paper is much clearer. The reviewers have only minor additional comments, which are included as comments in the attached PDF (please let our editorial office know if you have trouble accessing these comments).

Additionally:

The abstract and title still need some editing to reflect the actual research questions and findings. I suggest you emphasise knowledge of danger signs and care seeking practices, rather than the very general 'knowledge, attitudes, and practices.' The abstract completely leaves out care seeking for newborns, which seems to me that it would be the most likely variable to be impacted by increased knowledge of newborn danger signs. Currently, the focus is on ANC utilisation, which as I have noted before, is more likely to be a driver than an impact of increased knowledge.

Please add to the limitations the potential bias of only speaking to married man (and not women's unmarried partners). The difference in recruitment methods between men and women is explained well; however, it is still not clear why they completed different questionnaires and what the potential implications are.

Reviewers' comments:

Reviewer's Responses to Questions

**Comments to the Author**

1. If the authors have adequately addressed your comments raised in a previous round of review and you feel that this manuscript is now acceptable for publication, you may indicate that here to bypass the “Comments to the Author” section, enter your conflict of interest statement in the “Confidential to Editor” section, and submit your "Accept" recommendation.

Reviewer #2: All comments have been addressed

Reviewer #3: All comments have been addressed

2. Is the manuscript technically sound, and do the data support the conclusions?

Reviewer #2: Partly

Reviewer #3: Yes

3. Has the statistical analysis been performed appropriately and rigorously? 

Reviewer #2: Yes

Reviewer #3: Yes

4. Have the authors made all data underlying the findings in their manuscript fully available?

Reviewer #2: Yes

Reviewer #3: Yes

5. Is the manuscript presented in an intelligible fashion and written in standard English?

Reviewer #2: Yes

Reviewer #3: Yes

6. Review Comments to the Author

Reviewer #2: (No Response)

Reviewer #3: The authors have addressed most of the comments raised earlier. I have added a few additional comments in areas where there is still need for justification or more specific details.

Otherwise I recommend the article be accepted for publication after the minor corrections.

7. PLOS authors have the option to publish the peer review history of their article (what does this mean?). If published, this will include your full peer review and any attached files.

Reviewer #2: No

Reviewer #3: No

---

## [Author Response · Author response to Decision Letter 1]

22 Jan 2021

Second response to reviewers

PONE-D-20-06532R1

Men’s and women’s knowledge, attitudes and practices relating to pregnancy and neonatal care: a cross sectional study from Bungoma County, Kenya

PLOS ONE

Editor’s comments and responses

The abstract and title still need some editing to reflect the actual research questions and findings. I suggest you emphasise knowledge of danger signs and care seeking practices, rather than the very general 'knowledge, attitudes, and practices.' The abstract completely leaves out care seeking for newborns, which seems to me that it would be the most likely variable to be impacted by increased knowledge of newborn danger signs. Currently, the focus is on ANC utilisation, which as I have noted before, is more likely to be a driver than an impact of increased knowledge.

Thank you! The title and abstract have been revised to emphasize the study’s focus on neonatal and postnatal care-seeking, and remove the implication that pregnancy care utilization is an outcome.

Please add to the limitations the potential bias of only speaking to married man (and not women's unmarried partners). The difference in recruitment methods between men and women is explained well; however, it is still not clear why they completed different questionnaires and what the potential implications are.

Thank you. The Limitations section has been revised to also include the bias of only speaking to married men, and to note the study logistics that determined the use of different questionnaires; also noting that this difference constrains comparability between men’s and women’s answers.

Reviewers comments in body of PDF and responses (line numbers refer to PDF)

Line 111: Reviewer comment: “what about attitudes?”

Response: The title has been altered to no longer refer to attitudes.

Line 118: Reviewer comment: “This section should be written to have relevance to the study subject matter. For example, demonstrate how agricultural activities are related to men and women's KAP related to pregnancy and neonatal care. How is the population of the county related to the studied population when this is more of a hospital-based study?”

Response: The text has been revised to note the ecological relevance.

Line 127: Reviewer: how were these sample sizes determined/ calculated?

Response: the convenience sampling has been emphasized in this sentence and below.

Line 128: Reviewer: state clearly the one year period- between which actual months that constitute the year. This must be known to inform how screening/recruitment was done.

Response: the period has been specified more accurately

Line 129: Reviewer: What is the justification for use of this method? this is the poorest method of sampling. it is also inappropriate for sample size and statistical tools are intended for use to determine associations and aimed at generalization.

Response: We have added text to note the sampling was determined by the resources available to the project team. With this disclosure to inform readers, we believe the subsequent analysis and interpretation is proportionate and holds value.

Line 130: Reviewer: Reviewer: specify the market centres where recruitment was done- how many and which ones? what is their vicinity in relation to the hospitals?

Response: Their relationship to the hospitals in the broader study has been specified.

Line 130: Reviewer: be consistent in use of terms- 'women' instead of 'mothers', in line with your study title.

Response: we have changed ‘mother’ to ‘women’ or equivalent, throughout, except where needed to specify the parental role.

Line 130: Reviewer: is it possible that another reason for the selection of these hospitals is because the intervention- the collaborative newborn support project- had taken place at these facilities?

Response: we have added this detail.

Line 338: Reviewer: the sampling procedure used- convenience this is not only the weakest type of procedure, but it is also inappropriate for a study sample that is calculated using methods aimed at representativeness.

Response: we have strengthened our acknowledgement of this limitation

Line 353: Reviewer: The findings can only be generalized to the target population attending ANC, stretching it to the County when it is a hospital-based study, and using convenience sampling method is misleading.

Response: We have reworded to avoid the over-generalisation

---

## [Editor Report · Decision Letter 2]

29 Apr 2021

Men’s and women’s knowledge of danger signs relevant to postnatal and neonatal care-seeking: a cross sectional study from Bungoma County, Kenya

PONE-D-20-06532R2

Dear Dr. Gitaka,

We’re pleased to inform you that your manuscript has been judged scientifically suitable for publication and will be formally accepted for publication once it meets all outstanding technical requirements.

Kind regards,

Tanya Doherty, PhD

Academic Editor

PLOS ONE
---

## [Editor Report · Acceptance letter]

4 May 2021

PONE-D-20-06532R2 

Men’s and women’s knowledge of danger signs relevant to postnatal and neonatal care-seeking: a cross sectional study from Bungoma County, Kenya 

Dear Dr. Gitaka:

I'm pleased to inform you that your manuscript has been deemed suitable for publication in PLOS ONE. Congratulations! Your manuscript is now with our production department. 

Kind regards, 

on behalf of

Professor Tanya Doherty 

Academic Editor

PLOS ONE